# FAST RoPE ATTENTION: COMBINING THE POLYNOMIAL METHOD AND FAST FOURIER TRANSFORM

## ABSTRACT

The transformer architecture has been widely applied to many machine learning tasks. A main bottleneck in the time to perform transformer computations is a task called attention computation. [Alman and Song, NeurIPS 2023] have shown that in the bounded entry regime, there is an almost linear time algorithm to approximate the attention computation. They also proved that the bounded entry assumption is necessary for a fast algorithm assuming the popular Strong Exponential Time Hypothesis.

A new version of transformer which uses position embeddings has recently been very successful. At a high level, position embedding enables the model to capture the correlations between tokens while taking into account their position in the sequence. Perhaps the most popular and effective version is Rotary Position Embedding (RoPE), which was proposed by [Su, Lu, Pan, Murtadha, Wen, and Liu, Neurocomputing 2024].

A main downside of RoPE is that it complicates the attention computation problem, so that previous techniques for designing almost linear time algorithms no longer seem to work. In this paper, we show how to overcome this issue, and give a new algorithm to compute the RoPE attention in almost linear time in the bounded entry regime. (Again, known lower bounds imply that bounded entries are necessary.) Our new algorithm combines two techniques in a novel way: the polynomial method, which was used in prior fast attention algorithms, and the Fast Fourier Transform.

## 1 INTRODUCTION

Large language models (LLMs) are among the most impactful tools in modern machine learning. LLMs such as Transformer (Vaswani et al., 2017), BERT (Devlin et al., 2018), GPT-3 (Brown et al., 2020), PaLM (Chowdhery et al., 2022), OPT (Zhang et al., 2022a), GPT-4 (Achiam et al., 2023), Gemini (Team et al., 2023), Gemini 1.5 (Reid et al., 2024), Claude3 (Anthropic, 2024), GPT-4o (OpenAI, 2024a), o1 (OpenAI, 2024b), can process natural language more effectively than smaller models or traditional algorithms. This means that they can understand and generate more complex and nuanced language, which can be useful for a variety of tasks such as language translation, question answering, and sentiment analysis. LLMs can also be adapted to multiple purposes without needing to be retained from scratch.

**Attention Computation.** LLMs currently require massive time and computing resources to perform at scale. The major bottleneck to speeding up LLM operations is the time to perform a certain operation called an attention matrix computation (Vaswani et al., 2017; Radford et al., 2018; Devlin et al., 2018; Radford et al., 2019; Brown et al., 2020; Wang et al., 2020; Kitaev et al., 2020). These computations ask us to multiply the attention matrix $A$ with another value token matrix $V \in \mathbb{R}^{n \times d}$. More precisely, given three matrices $Q, K, V \in \mathbb{R}^{n \times d}$ (the query, key, and value token matrices), the goal is to output (an approximation of) the $n \times d$ matrix $\mathsf{Att}(Q, K, V)$ defined by $\mathsf{Att}(Q, K, V) := D^{-1}AV$ where the attention matrix $A \in \mathbb{R}^{n \times n}$ and diagonal matrix $D \in \mathbb{R}^{n \times n}$ are defined as $A := \exp(QK^\top/d)$ (with $\exp$ applied entry-wise), and $D := \mathrm{diag}(A\mathbf{1}_n)$. Here, $n$ is the input sequence length, and $d$ is the embedding dimension of the model, and one typically considers $d \ll n$ like $d = \Theta(\log n)$ in the time-intensive case of modeling long sequences.

The straightforward algorithm for this problem runs in roughly quadratic time. Moreover, there are known complexity-theoretic lower bounds (Keles et al., 2023; Alman & Song, 2023) proving that the problem cannot be solved in truly subquadratic time in the case when the input matrices $Q, K, V$ have large entries, assuming a popular conjecture from fine-grained complexity theory called the Strong Exponential Time Hypothesis (SETH (Impagliazzo & Paturi, 2001)) which we discuss more shortly.

In order to circumvent this lower bound, and inspired by the fact that the entries of the input matrices are typically bounded in realistic inputs (Zafrir et al., 2019; Katharopoulos et al., 2020b), a recent faster, almost linear-time algorithm (Alman & Song, 2023) was given, assuming that $\|Q\|_\infty, \|K\|_\infty, \|V\|_\infty$ are all bounded. Here the $\ell_\infty$-norm is given by $\|Q\|_\infty := \max_{i,j} |Q_{i,j}|$. Rather than explicitly compute all the entries of the attention matrix $A$, (Alman & Song, 2023) only *implicitly* uses it, by using an algorithmic tool called the *polynomial method*.

More precisely, they present two results, showing that when $d = O(\log n)$, and $B$ is the bound $\|Q\|_\infty, \|K\|_\infty, \|V\|_\infty \le B$, there is a sharp transition in the difficulty of attention computation at $B = \Theta(\sqrt{\log n})$. Here, $\|Q\|_\infty := \max_{i,j} |Q_{i,j}|$. First, if $B = o(\sqrt{\log n})$, then there is an $n^{1+o(1)}$ time algorithm to approximate $\mathrm{Att}(Q, K, V)$ up to $1/\mathrm{poly}(n)$ additive error. Second, if $B = \Theta(\sqrt{\log n})$, then assuming SETH, it is impossible to approximate $\mathrm{Att}(Q, K, V)$ up to $1/\mathrm{poly}(n)$ additive error in truly subquadratic time $n^{2-\Omega(1)}$. In other words, if $B = o(\sqrt{\log n})$, then the polynomial method gives an almost linear-time algorithm, and if $B$ is any bigger, then it is *impossible* to design an algorithm that substantially improves on the trivial quadratic time algorithm, no matter what algorithmic techniques one uses.

**Bounded entries in practice.** The theoretical results of (Alman & Song, 2023) offer an explanation for a phenomenon commonly observed in practice: attention computation becomes significantly more efficient when the input matrices have bounded entries. Indeed, a long line of work on LLM implementations has achieved speedups by combining bounds on weights with algorithmic techniques like quantization and low-degree polynomial approximation. For some examples, see (Zafrir et al., 2019; Katharopoulos et al., 2020b; Frantar et al., 2022; Perez et al., 2023; Dettmers et al., 2023; Egashira et al., 2024; Liu et al., 2024b; Xu et al., 2024b; Lin et al., 2025; Chen et al., 2025b; Liu et al., 2025; Ouyang et al., 2025; Deng et al., 2025; Hu et al., 2025c; Fu et al., 2025; Hu et al., 2025b; Park et al., 2025; Zeng et al., 2025; Yu et al., 2025; Wei et al., 2025).

**RoPE: Rotary Position Embedding.** This work mainly explores the efficient computation of an emerging type of attention, namely RoPE attention, which enables improved attention expressiveness while resulting in a more difficult computational problem. This property makes the efficient computation of RoPE attention more challenging, since a wide range of previous works (e.g., the algorithm in (Alman & Song, 2023)) cannot be applied in this new setting. Various industrial LLMs have adopted RoPE attention as their model components, making RoPE a standard approach in attention computation. Examples include Meta's open-source Llama family models (Touvron et al., 2023a;b; AI, 2024), Anthropic's private commercial model Claude 3 (Anthropic, 2024), and Apple's LLM architecture (McKinzie et al., 2024; Gunter et al., 2024) [1].

The inherent intuition of RoPE is to enhance the attention expressivity via rotating the queries and keys. Specifically, the rotation depends on the sequence positions, thereby ensuring that the inner product of vectors with position encoding can express the actual relative positions. Instantiation of RoPE attention is based on the $R_{j-i}$ matrices, which we will define below. These matrices perform position-aware rotations to the embeddings, which makes token pairs with smaller relative distances have larger correlations.

We now briefly describe the mathematical definition of the RoPE method. We will make use of $2 \times 2$ rotation matrices, which for an angle of rotation $\theta$, can be written as

$$R(\theta) := \begin{bmatrix} \cos\theta & -\sin\theta \\ \sin\theta & \cos\theta \end{bmatrix}.$$

As above, we denote the length of input sequences by $n$, and represent the dimension of embedding vectors by $d$. We assume here that $d$ is even.

---

[1] The use of RoPE attention can be found in the technical reports of these LLMs. See page 3 of (Touvron et al., 2023a), page 5 of (Touvron et al., 2023b), page 7 of (Llama Team, 2024), and page 3 of (Gunter et al., 2024).

For $i, j \in [n]$, we now define the overall relative rotation matrix for tokens at positions $j$ and $i$, which we denote by $R_{j-i} \in \mathbb{R}^{d \times d}$. As indicated by the notation, it depends only on the difference $j - i$. $R_{j-i}$ is defined as a diagonal block matrix with $d/2$ blocks of size $2 \times 2$ along the diagonal, given by

$$
R_{j-i} = \begin{bmatrix} R((j-i)\theta_1) & 0 & \cdots & 0 \\ 0 & R((j-i)\theta_2) & \cdots & 0 \\ \vdots & \vdots & \ddots & \vdots \\ 0 & 0 & \cdots & R((j-i)\theta_{d/2}) \end{bmatrix}.
$$

The angle frequencies are given by $\theta_k = \alpha^{-2(k-1)/d}$ for $k \in [d/2]$. Here one thinks of the angle $\alpha$ as a fixed constant for all $i$ and $j$; in the original RoPE it is $10^4$ (see details in Equation (15) in page 5 of (Su et al., 2024)).

These $R_{j-i}$ matrices are incorporated into attention as follows. Let $W_Q, W_K, W_V \in \mathbb{R}^{d \times d}$ denote the model parameters. Let $X \in \mathbb{R}^{n \times d}$ denote the latent representation of a sentence with length $n$. Then, for $i, j \in [n]$, a new attention matrix can be defined as

$$
A_{i,j} := \exp(\underbrace{X_{i,*}}_{1 \times d} \underbrace{W_Q}_{d \times d} \underbrace{R_{j-i}}_{d \times d} \underbrace{W_K^\top}_{d \times d} \underbrace{X_{j,*}^\top}_{d \times 1}). \tag{1}
$$

As in the usual attention mechanism, the final goal is to output an $n \times d$ size matrix $D^{-1}AXW_V$ where $D := \mathrm{diag}(A\mathbf{1}_n) \in \mathbb{R}^{n \times n}$.

**Formulation of RoPE Attention.** In this paper, we give a new algorithm for RoPE attention. We now formally define the problem we will solve. Notably, our algorithm actually solves the following *generalization* of RoPE attention, which captures RoPE (as we described it above) as well as many natural variants on RoPE that future work may want to consider. We emphasize that changing the many parameters which go into the RoPE definition would still be captured by our generalization below.

**Definition 1.1** (A General Approximate RoPE Attention Computation, ARAttC). *Let $\epsilon > 0$ denote an accuracy parameter, and $B > 0$ denote a magnitude parameter. We define $S$ as $S \subseteq [d] \times [d]$ and $|S| = O(d)$. Given a set of matrices $W_{-(n-1)}, \cdots W_{-1}, W_0, W_1, \cdots W_{n-1} \in \mathbb{R}^{d \times d}$ where $\mathrm{supp}(W_i) \subset S$ for all $i \in \{-(n-1), \cdots, -1, 0, 1, \cdots, n-1\}$. Given $Q \in \mathbb{R}^{n \times d}$, $K \in \mathbb{R}^{n \times d}$, and $V \in \mathbb{R}^{n \times d}$ with the guarantee that $\|Q\|_\infty, \|K\|_\infty, \|V\|_\infty \leq B$ and $\|W\|_\infty \leq 1$. We define matrix $A \in \mathbb{R}^{n \times n}$ as, for $i, j \in [n]$,*

$$
A_{i,j} := \exp(\underbrace{Q_{i,*}}_{1 \times d} \underbrace{W_{i-j}}_{d \times d} \underbrace{K_{j,*}^\top / \sqrt{d}}_{d \times 1}), \forall i \in [n], j \in [n]
$$

*We let $D := \mathrm{diag}(A\mathbf{1}_n)$ and $\mathsf{ARAttC} := D^{-1}AV$. The goal of General Approximate RoPE Attention Computation is to output a matrix $T \in \mathbb{R}^{n \times d}$ such that $\|T - \mathsf{ARAttC}\|_\infty \leq \epsilon$.*

**Remark 1.2.** *RoPE attention as defined above (Eq. (1)) corresponds to this problem where we restrict each of the matrices $W_i \in \mathbb{R}^{d \times d}$ for all $i \in \{-(n-1), , \cdots, -1, 0, 1, \cdots, n-1\}$ in Definition 1.1 to be diagonal block matrices, where each matrix has $d/2$ blocks and each block has size $2 \times 2$. Note that the $1/d$ factor inside $\exp$ in the definition of $A$ is a normalization factor.*

**Our Results.**

Our main result is a new algorithm which computes General Approximate RoPE Attention Computation in almost linear time:

**Theorem 1.3** (main result, upper bound). *Suppose $\epsilon = 1/\mathrm{poly}(n)$, $B = o(\sqrt{\log n})$, and $d = O(\log n)$. There is an $n^{1+o(1)}$ time algorithm to approximate $\mathsf{ARAttC}$ up to $\epsilon$ additive error.*

In other words, although RoPE attention is more complicated than the usual attention, we are able to achieve the same running time for this more expressive version. This is, to our knowledge, the first fast algorithm for RoPE attention with provable guarantees. As we will discuss more shortly, there is a substantial barrier to using prior algorithmic techniques for attention in the setting of RoPE

attention, and we overcome this barrier using a novel approach combining the polynomial method with Fast Fourier transforms.

Furthermore, we prove that the bound of $B = o(\sqrt{\log n})$ used by our algorithm is necessary, since when $B$ is any bigger, it is impossible to design a truly subquadratic time algorithm:

**Theorem 1.4** (main result, lower bound). *Assuming* SETH*, for every $q > 0$, there are constants $C, C_a, C_b > 0$ such that: there is no $O(n^{2-q})$ time algorithm for the problem* ARAttC$(n, d = C \log n, B = C_b\sqrt{\log n}, \epsilon = n^{-C_a})$.

To emphasize, our Theorem 1.4 doesn't just prove that our algorithmic approach cannot give a nontrivial algorithm when $B = \Omega(\sqrt{\log n})$, but more generally that it is impossible to design a nontrivial algorithm, no matter what algorithmic techniques one uses.

Our Theorem 1.4 closely matches the parameters of prior lower bounds on the usual attention problem (and it is not too difficult to prove given these prior lower bounds). Because of the increased complexity of RoPE attention, it previously seemed conceivable that one could prove a stronger lower bound for ARAttC; perhaps surprisingly, our Theorem 1.3 shows that it is actually tight. Since the proof of Theorem 1.4 is so similar to prior work, we provide it in Section B.2 in the Appendix.

**Technique Overview: Limitation of Prior Techniques**

Prior fast algorithms with provable guarantees for attention are critically based on an algorithmic technique called the *polynomial method* (Alman & Song, 2023; 2024a;b). This is a technique for finding low-rank approximations of certain structured matrices. More precisely, suppose $M \in \mathbb{R}^{n \times n}$ is a low-rank matrix, and $f : \mathbb{R} \to \mathbb{R}$ is any function. Let $f(M)$ denote the matrix where $f$ is applied entry-wise to $M$. In general, although $M$ is low-rank, the matrix $f(M)$ may be a full-rank matrix. However, the polynomial method says that if $f$ can be approximated well by a low-degree polynomial, then $f(M)$ can be approximated well by a low-rank matrix. Since the usual attention matrix is defined by applying $\exp$ entry-wise to a low-rank matrix, prior algorithms approximate $\exp$ with a polynomial, then uses the polynomial method to approximate the attention matrix with a low-rank matrix which can be used to quickly perform the necessary linear-algebraic operations.

Although this approach has been successful in prior work on designing faster algorithms for many problems related to attention, it fundamentally cannot apply to RoPE attention. The key issue is that in RoPE attention, the underlying matrix which $\exp$ is applied to no longer needs to have low rank. Indeed, let $A$ denote the RoPE attention matrix (defined in Equation (1) above) and let $M$ denote $A$ before it was entry-wise exponentiated. Even in the simplest case $d = 1$, one can see that by picking the $R_{j-i}$ entries appropriately (and the entries of all other matrices in Equation (1) to equal 1), one can choose $M$ to be *any* Toeplitz matrix (i.e., matrix whose $(i, j)$ entry depends only on the difference $j - i$). The polynomial method then cannot be used to argue that $A$ is approximately low-rank, since $M$ itself is not low-rank.

**Technique Overview: Combining the Polynomial Method and Fast Fourier Transform**

Although Toeplitz matrices are typically not low-rank matrices, there is a vast literature on algorithms for manipulating them using the Fast Fourier transform. (The reader may be more familiar with this fact for circulant matrices; this same algorithm can be applied by first embedding the Toeplitz matrix into a circulant matrix with twice the side-length.) Notably, it is not hard to notice that applying *any* function entry-wise to a Toeplitz matrix results in another Toeplitz matrix, so if $M$ were indeed a Toeplitz matrix as described in the previous paragraph, one could use the Fast Fourier transform to perform operations with the resulting matrix $A$.

However, even in the case of $d = 1$, the matrix $M$ can actually be a more general type of matrix which we call a *rescaled Toeplitz matrix* (because of the $X$ matrices in Equation (1)). This is a matrix of the form $D_1 C D_2$ for diagonal matrices $D_1, D_2$ and Toeplitz matrix $C$. Unfortunately, applying a function entry-wise to a rescaled Toeplitz matrix need not result in another rescaled Toeplitz matrix.

Our main algorithmic idea is a new version of the polynomial method: we prove that if $M$ is a rescaled Toeplitz matrix, or even a sum of a small number of rescaled Toeplitz matrices, and one applies a function $f$ entry-wise to $M$ such that $f$ has a low-degree polynomial approximation, then the resulting matrix can be approximated by a sum of a relatively small number of rescaled Toeplitz matrices. In our case, we use this to write the RoPE attention matrix as a sum of rescaled Toeplitz

matrices, each of which is then manipulated using the Fast Fourier transform to yield our final algorithm.

We believe our new approach, of applying polynomial approximations entry-wise to structured matrices other than low-rank matrices, may be broadly applied in other settings as well. Although the polynomial method has been applied in many algorithmic contexts, to our knowledge, it was always previously used to find a low-rank approximation of the underlying matrix, and not another structured decomposition like this.

**Algorithmic techniques in practice.** We emphasize that our two core techniques, the polynomial method and Fast Fourier transform, are both prevalent in practice. The polynomial method is particularly used in numerous practical algorithms for attention (Banerjee et al., 2020; Keles et al., 2023; Zhang et al., 2024b). For example, see detailed discussions in (Zhang et al., 2024b). Our new algorithm improves on these approaches in part by using *theoretically optimal* polynomials for exponentials, and combining them with the Fast Fourier transform, to give provable guarantees about their correctness and near linear running time. To our knowledge, the Fast Fourier transform has not been used in this way in prior attention algorithms.

**Roadmap.** In Section 2, we present our related work. In Section 3, we define certain basic notations for linear algebra. In Section 4, we commence by solving the linear case. Finally, we provide a conclusion in Section 5.

## 2 RELATED WORK

**Polynomial Method for Attention.** (Alman & Song, 2023; 2024b) utilize polynomial kernel approximation techniques proposed by (Aggarwal & Alman, 2022) to speed up both training and inference of a single attention layer, achieving almost linear time complexity. This method is further applied to multi-layer transformer (Liang et al., 2024c), tensor attention (Alman & Song, 2024a; Liang et al., 2024e), LoRA (Hu et al., 2024b), Hopfield model (Hu et al., 2023; 2024a; Wu et al., 2024; Xu et al., 2024a), differentially private cross attention (Liang et al., 2024d), and Diffusion Transformer (Hu et al., 2024d; Shen et al., 2025a), adapters (Hu et al., 2022; Zhang et al., 2023a; Gao et al., 2023a; Shi et al., 2023a), calibration approaches (Zhao et al., 2021; Zhou et al., 2023), multitask fine-tuning strategies (Gao et al., 2021a; Xu et al., 2023b; Von Oswald et al., 2023; Xu et al., 2024c), prompt tuning techniques (Gao et al., 2021b; Lester et al., 2021), scratchpad approaches (Nye et al., 2021), instruction tuning methodologies (Li & Liang, 2021; Chung et al., 2022; Mishra et al., 2022), symbol tuning (Wei et al., 2023), black-box tuning (Sun et al., 2022), reinforcement learning from the human feedback (RLHF) (Ouyang et al., 2022), chain-of-thought reasoning (Wei et al., 2022; Khattab et al., 2022; Yao et al., 2023; Zheng et al., 2024) and various other strategies. We will also use the polynomials of (Aggarwal & Alman, 2022) here.

**Fast Fourier transform.** The Fast Fourier transform algorithm (Cooley & Tukey, 1965) can multiply the $n$ by $n$ Discrete Fourier transform matrix times an input vector in $O(n \log n)$ time. This algorithm is impactful in many areas, including image processing, audio processing, telecommunications, seismology, and polynomial multiplication. Due to its fundamental importance, a significant body of modern research has been dedicated to further accelerating the Fast Fourier transform. See Appendix E for an overview of some of the vast literature.

In particular, recent work (Fein-Ashley et al., 2025) has used the FFT for computing attention faster, showing that it can perform well compared to the hardware-accelerated matrix multiplication that is typically used.

**Other Algorithms for Computing Attention.** Due to its quadratic time complexity with respect to context length (Vaswani et al., 2017), the attention mechanism has faced criticism. To address this issue, various approaches have been employed to reduce computational overhead and improve scalability, including sparse attention (Child et al., 2019; Beltagy et al., 2020; Zaheer et al., 2020; Hubara et al., 2021; Kurtic et al., 2023; Frantar & Alistarh, 2023; Shi et al., 2023a; Deng et al., 2023a; Li et al., 2024c; Han et al., 2024a; Liang et al., 2024a), low-rank approximations (Razenshteyn et al., 2016; Li et al., 2016; Hu et al., 2022; Zeng & Lee, 2024; Hu et al., 2024b), and kernel-based methods (Charikar et al., 2020; Liu & Zenke, 2020; Deng et al., 2023b; Zandieh et al., 2023; Liang et al., 2024b).

Additionally, linear attention has emerged as a significant fast alternative to softmax attention, prompting substantial research in this area (Tsai et al., 2019; Katharopoulos et al., 2020a; Schlag et al., 2021; Zhang et al., 2023b; Sun et al., 2023; Ahn et al., 2024; Shi et al., 2023b; Zhang et al., 2024b; Deng et al., 2023c; Li et al., 2024a). Moreover, other related works examine various aspects of attention computation, including I/O complexity (Dao et al., 2022; Dao, 2023; Li et al., 2024d), circuit complexity (Chen et al., 2024c;a; Li et al., 2025a), differential privacy (Gao et al., 2024a; Liang et al., 2024d), weights pruning (Frantar & Alistarh, 2023; Sun et al., 2024; Shen et al., 2025c; Liang et al., 2025), half-space reporting (Jiang et al., 2021; Chen et al., 2024b), graph neural network (Qin et al., 2023; Chang et al., 2024), regression problems (Gao et al., 2023b), and quantum algorithms (Gao et al., 2023c; Zhao et al., 2024). A recent work (Alman & Song, 2025) has investigated the significance of selecting large weights in approximating attention computation to enhance expressiveness.

**Accelerated Computation in Machine Learning.** Due to the increasing scale of training data in various applications of machine learning, including but not limited to human language (Devlin et al., 2019), images (Awais et al., 2025), audio (Schneider et al., 2019), and social networks (Catanese et al., 2011), accelerated computation of modern ML models has been a central concern of today's AI community (Venkataramani et al., 2015; Bender et al., 2021; McDonald et al., 2022). Regression models have long been a simple yet effective solution to many ML problems, such as optimization (Bubeck, 2015), neural network training (Brand et al., 2021; Song et al., 2024b), and signal processing (Rabiner et al., 1978; Subrahmanya & Shin, 2009). A wide range of techniques has been applied to accelerate regression computation, such as pre-conditioning (Yang et al., 2018; Kelner et al., 2022; Song et al., 2024a) and sketching (Song & Yu, 2021; Reddy et al., 2022; Song et al., 2023c).

Diffusion models have recently become a fundamental game changer in content generation, producing realistic and aesthetically desirable images (Ho et al., 2020; Song et al., 2021b;a) and videos (Ho et al., 2022; Blattmann et al., 2023) that meet high standards. These successful stories also extend to many non-visual applications, such as text generation (Lin et al., 2023; Sahoo et al., 2024), drug discovery (Xu et al., 2023a; Wen et al., 2024), recommender systems (Wang et al., 2023; Yang et al., 2023), and time series forecasting (Tashiro et al., 2021; Rasul et al., 2021). A recent work (Liu et al., 2024a) has explored the intersection of diffusion models and socially aware recommender systems, aiming to mitigate the social heterophily effect through diffusion-based social information enhancement. Recent works have revealed that some specific types of diffusion modes can be approximated in almost linear time with provably efficient criteria (Hu et al., 2024d;c; 2025a; Gong et al., 2025). To accelerate the inference and training of diffusion models, enabling real-time content generation for users and fast model updates for model owners, recent progress includes shortcut models (Dao et al., 2024; Frans et al., 2024; Chen et al., 2025a), pre-conditioning (Garber & Tirer, 2024; Ma et al., 2025), lazy learning (Nitzan et al., 2024; Shen et al., 2025b), and weight pruning (Ma et al., 2024; Castells et al., 2024). Graph neural networks (GNNs) are essential tools for modeling relational data (Kipf & Welling, 2016; Wu et al., 2019; Demirel et al., 2022), powering a wide range of applications, including traffic forecasting (Diao et al., 2019; Shao et al., 2022; Han et al., 2024b), fake news detection (Xu et al., 2022; Chang et al., 2024), social network analysis (Fan et al., 2019; Zhang et al., 2022b), human action recognition (Peng et al., 2020; Li et al., 2021; Fu et al., 2021), and e-commerce (Ying et al., 2018; He et al., 2020). Recent advances in acceleration include model quantization (Tailor et al., 2021; Liu et al., 2023), lazy learning (Narayanan et al., 2022; Xue et al., 2024), and sketching (Ding et al., 2022; Chamberlain et al., 2023). A recent study (Zhang et al., 2024a) accelerated GNNs using both lazy propagation and variance-reduced random sampling of finite sums, resulting in a linear-time GNN with broad applications in e-commerce.

## 3 PRELIMINARIES

In Section 3.1, we define several notations. We discuss some backgrounds for fast circulant transform. In Section 3.2, we provide a tool from previous work about how to control error by using low-degree polynomial to approximate exponential function. In Section 3.3, we discuss some backgrounds about fast circulant transform. In Section 3.4, we formalize the toeplitz matrix and introduce the tools we will use. In Section 3.5, we define rescaled circulant matrix and provide some basic tools for it.

## 3.1 NOTATION

For nonnegative integer $n$, we use $[n]$ to denote set $\{1, 2, \cdots, n\}$. We say $O(n \log n)$ is nearly-linear time. We say $O(n^{1+o(1)})$ is almost linear time (We prove folklore fact for explaining the connection between nearly-linear and almost-linear). For a vector $a$, we represent the diagonal matrix where the $(i, i)$-th entry is $a_i$ with $\mathrm{diag}(a)$. We use $\mathrm{supp}$ to denote the support of a matrix, i.e., the set of entries where the matrix is nonzero. For a matrix $A$, we use $A^\top$ to denote transpose of $A$. Given two vectors $a, b$ of the same length, we use $a \circ b$ to denote their entry-wise product, i.e., the vector where the $i$-th entry is $a_i b_i$. Given two matrices $A, B$ of the same dimensions, we similarly use $A \circ B$ to denote their entry-wise Hadamard product, i.e., the matrix where the $(i, j)$-th entry is $A_{i,j} B_{i,j}$. For a non-negative integer $t$ and a matrix $A$, we use $A^{\circ t} := \underbrace{A \circ A \circ \cdots \circ A}_{t \text{ terms}}$, i.e., $(A^{\circ t})_{i,j} = A_{i,j}^t$.

## 3.2 POLYNOMIAL APPROXIMATION OF EXPONENTIAL

To control the error dependence of our proposed approximate algorithm, we present a standard technical lemma used in many previous works (Alman & Song, 2023; 2024a;b).

**Lemma 3.1** ((Aggarwal & Alman, 2022)). *Let $\epsilon \in (0, 0.1)$ and $B > 1$. There is a polynomial $P : \mathbb{R} \to \mathbb{R}$ of degree $g := \Theta\left(\max\left\{\frac{\log(1/\epsilon)}{\log(\log(1/\epsilon)/B)}, B\right\}\right)$ such that for all $x \in [0, B]$, we have*

$$|P(x) - \exp(x)| < \epsilon.$$

*Furthermore, $P$ can be computed efficiently: its coefficients are rational numbers with $\mathrm{poly}(g)$-bit integer numerators and denominators which can be computed in $\mathrm{poly}(g)$ time.*

## 3.3 FAST CIRCULANT TRANSFORM

Circulant matrices have been widely used in applied mathematics (Meckes, 2009; Adamczak, 2010), compressive sensing (Rauhut et al., 2012; Krahmer et al., 2014; Nelson et al., 2014) and regression literature (Song et al., 2023b). Here we provide the formal definition.

**Definition 3.2** (Circulant matrix). *Let $a \in \mathbb{R}^n$ denote a length-$n$ vector. We define $\mathsf{Circ} : \mathbb{R}^n \to \mathbb{R}^{n \times n}$ as,*

$$\mathsf{Circ}(a) := \begin{bmatrix} a_1 & a_n & a_{n-1} & \cdots & a_2 \\ a_2 & a_1 & a_n & \cdots & a_3 \\ a_3 & a_2 & a_1 & \cdots & a_4 \\ \vdots & \vdots & \vdots & \ddots & \vdots \\ a_n & a_{n-1} & a_{n-2} & \cdots & a_1 \end{bmatrix}.$$

**Fact 3.3** ((Gray et al., 2006)). *Let $a \in \mathbb{R}^n$ represent a length-$n$ vector. We define $\mathsf{Circ}$ as Definition 3.2. Let $F \in \mathbb{C}^{n \times n}$ be the discrete Fourier transform matrix. By leveraging the property of discrete Fourier transform, we have*

$$\mathsf{Circ}(a) = F^{-1} \mathrm{diag}(Fa) F.$$

*Thus, we can multiply $\mathsf{Circ}(a)$ with an input vector of length $n$ in $O(n \log n)$ time using the Fast Fourier transform algorithm.*

## 3.4 TOEPLITZ MATRIX

The Toeplitz matrix is similar to a circulant matrix, but is defined through a vector in $\mathbb{R}^{2n-1}$. Both matrices exhibit identical time complexity when performing a matrix-vector product.

**Definition 3.4** (Toeplitz matrix). *Let $a := (a_{-(n-1)}, \cdots, a_{-1}, a_0, a_1, \cdots, a_{n-1}) \in \mathbb{R}^{2n-1}$ denote a length-$(2n-1)$ vector. We define $\mathsf{Toep} : \mathbb{R}^{2n-1} \to \mathbb{R}^{n \times n}$ as*

$$\mathsf{Toep}(a) := \begin{bmatrix} a_0 & a_{-1} & a_{-2} & \cdots & a_{-(n-1)} \\ a_1 & a_0 & a_{-1} & \cdots & a_{-(n-2)} \\ a_2 & a_1 & a_0 & \cdots & a_{-(n-3)} \\ \vdots & \vdots & \vdots & \ddots & \vdots \\ a_{(n-1)} & a_{(n-2)} & a_{(n-3)} & \cdots & a_0 \end{bmatrix}.$$

*In other words, $\mathsf{Toep}(a)_{i,j} := a_{i-j}$.*

**Fact 3.5** (Fact B.7 in (Liang et al., 2024a)). *We define* Toep *as Definition 3.4, and define* Circ *as Definition 3.2. Given a length-$(2n-1)$ vector $a \in \mathbb{R}^{2n-1}$ (for convenience, we use $a_i \in \mathbb{R}$ to denote the entry of vector where $i \in \{-(n-1), -(n-2), \cdots, 0, \cdots, (n-2), (n-1)\}$). Let $a' \in \mathbb{R}^{2n}$, such that $a' = [a_0, a_1, \ldots, a_{n-1}, 0, a_{-(n-1)}, \ldots, a_{-1}]^\top$. For any $x \in \mathbb{R}^n$, we have*

$$\mathsf{Circ}(a') \begin{bmatrix} x \\ \mathbf{0}_n \end{bmatrix} = \begin{bmatrix} \mathsf{Toep}(a) & \mathsf{Resi}(a) \\ \mathsf{Resi}(a) & \mathsf{Toep}(a) \end{bmatrix} \cdot \begin{bmatrix} x \\ \mathbf{0}_n \end{bmatrix} = \begin{bmatrix} \mathsf{Toep}(a)x \\ \mathsf{Resi}(a)x \end{bmatrix},$$

*where the residual matrix is defined as*

$$\mathsf{Resi}(a) := \begin{bmatrix} 0 & a_{n-1} & a_{n-2} & \cdots & a_2 & a_1 \\ a_{-(n-1)} & 0 & a_{n-1} & \cdots & a_3 & a_2 \\ a_{-(n-2)} & a_{-(n-1)} & 0 & \cdots & a_4 & a_3 \\ \vdots & \vdots & \vdots & \ddots & \vdots & \vdots \\ a_{-2} & a_{-3} & a_{-4} & \cdots & 0 & a_{n-1} \\ a_{-1} & a_{-2} & a_{-3} & \cdots & a_{-(n-1)} & 0 \end{bmatrix}.$$

**Remark 3.6.** *Facts 3.3 and 3.5 imply that the matrix-vector product of a Toeplitz matrix can be computed in $O(n \log n)$ time.*

### 3.5 RESCALED TOEPLITZ MATRIX

Our algorithm will critically involve manipulating a certain kind of structured matrix we call a *rescaled Toeplitz matrix*. In this section we define these matrices and prove basic properties which we will use.

**Definition 3.7** (Rescaled Toeplitz Matrix). *We say a square matrix $M \in \mathbb{R}^{n \times n}$ is* rescaled Toeplitz *if there are diagonal matrices $D_1, D_2 \in \mathbb{R}^{n \times n}$ and a Toeplitz matrix $C \in \mathbb{R}^{n \times n}$ such that $M = D_1 C D_2$.*

**Fact 3.8.** *If $M \in \mathbb{R}^{n \times n}$ is a rescaled Toeplitz matrix (see Definition 3.7), then given as input a vector $v \in \mathbb{R}$, one can compute the matrix-vector product $Mv$ in $O(n \log n)$ time.*

*Proof.* Suppose $M = D_1 C D_2$, we first compute $D_2 v$ straightforwardly in $O(n)$ time. Then we compute $C \cdot (D_2 v)$ in $O(n \log n)$ time. Finally, we compute $D_1 \cdot (C D_2 v)$ in $O(n)$ time. $\qquad\square$

**Lemma 3.9.** *If $A$ and $B$ are rescaled Toeplitz matrices, then $A \circ B$ is also a rescaled Toeplitz matrix.*

*Proof.* Suppose $A = \mathrm{diag}(a_1) A_2 \mathrm{diag}(a_3)$ where $A_2$ is a Toeplitz matrix, and $B = \mathrm{diag}(b_1) B_2 \mathrm{diag}(b_3)$ where $B_2$ is a Toeplitz matrix. We can show

$$A \circ B = (\mathrm{diag}(a_1) A_2 \mathrm{diag}(a_3)) \circ (\mathrm{diag}(b_1) B_2 \mathrm{diag}(b_3))$$
$$= \mathrm{diag}(a_1) \mathrm{diag}(b_1)((A_2 \mathrm{diag}(a_3)) \circ (B_2 \mathrm{diag}(b_3)))$$
$$= \mathrm{diag}(a_1) \mathrm{diag}(b_1)(A_2 \circ B_2) \mathrm{diag}(a_3) \mathrm{diag}(b_3).$$

Therefore, we know $A \circ B$ is also a rescaled Toeplitz matrix. $\qquad\square$

**Lemma 3.10.** *If $A_1, \cdots, A_t$ are rescaled Toeplitz matrices, then for any vector $v$, we have $(A_1 \circ A_2 \circ \cdots \circ A_t)v$ can be computed in $O(tn \log n)$ time.*

*Proof.* The proof directly follows from applying Lemma 3.9 and Fact 3.8, $t$ times. $\qquad\square$

## 4 HOW TO COMPUTE THE LINEAR ATTENTION UNDER ROPE

Before starting to work on RoPE softmax attention, here we consider the simpler problem of computing RoPE *linear* attention. This linear attention does not have entry-wise $\exp$.

**Definition 4.1** (Linear Attention). *Let $S \subseteq [d] \times [d]$ denote a support and $|S| = O(d)$. Given $W_{-(n-1)}, \cdots W_{-1}, W_0, W_1, \cdots W_{n-1} \in \mathbb{R}^{d \times d}$ and for all $i \in \{-(n-1), \cdots, -1, 0, 1, \cdots, n-1\}$. Given $Q \in \mathbb{R}^{n \times d}$ and $K \in \mathbb{R}^{n \times d}$, $V \in \mathbb{R}^{n \times d}$. We define matrix $A \in \mathbb{R}^{n \times n}$ such as follows*

$$A_{i,j} := (\underbrace{Q_{i,*}}_{1 \times d} \underbrace{W_{i-j}}_{d \times d} \underbrace{K_{j,*}^\top}_{d \times 1}), \forall i \in [n], j \in [n]$$

*We define $D := \text{diag}(A\mathbf{1}_n)$. The attention computation is going to output an $n \times d$ matrix $D^{-1}AV$.*

For this linear version, we now show how to reduce it to $O(|S|)$ Fast Fourier transforms (FFTs), each of which can be performed in $O(n \log n)$ time. Intuitively, our algorithm is going to write $A \in \mathbb{R}^{n \times n}$ in the form $A = \sum_{(l_1, l_2) \in S} B_{l_1, l_2}$ where each $B_{l_1, l_2} \in \mathbb{R}^{n \times n}$ is a rescaled Toeplitz matrix.

Recall the support $S$:

**Definition 4.2.** *Given a collection of weight matrices $W_{-(n-1)}, \cdots W_{-1}, W_0, W_1, \cdots W_{n-1}$, we use $S$ to denote their support such that $\forall i \in \{-(n-1), \cdots, n-1\}, \text{supp}(W_i) = S$.*

**Definition 4.3** (one-sparse matrix). *For each pair $(\ell_1, \ell_2) \in S$, and $i, j \in [n]$, define the matrix $W_{i-j}^{\ell_1, \ell_2} \in \mathbb{R}^{d \times d}$ to be all 0s except that entry $(\ell_1, \ell_2)$ is equal to $(W_{i-j})_{\ell_1, \ell_2}$.*

**Claim 4.4.** *Let one sparse matrix $W_{i-j}^{\ell_1, \ell_2} \in \mathbb{R}^{d \times d}$ be defined as Definition 4.3. Then,*

$$W_{i-j} = \sum_{(\ell_1, \ell_2) \in S} W_{i-j}^{\ell_1, \ell_2}.$$

*Proof.* We can show that

$$W_{i-j} = \sum_{(\ell_1, \ell_2) \in S} \underbrace{e_{\ell_1}}_{d \times 1} \underbrace{(W_{i-j})_{\ell_1, \ell_2}}_{\text{scalar}} \underbrace{e_{\ell_2}^\top}_{1 \times d} = \sum_{(\ell_1, \ell_2) \in S} W_{i-j}^{\ell_1, \ell_2}$$

where the second step follows from Definition 4.3. $\qquad\square$

**Definition 4.5.** *For each pair $(\ell_1, \ell_2) \in S$, we define matrix $A^{\ell_1, \ell_2} \in \mathbb{R}^{n \times n}$ as follows:*

$$A_{i,j}^{\ell_1, \ell_2} := \underbrace{Q_{i,*}}_{1 \times d} \underbrace{W_{i-j}^{\ell_1, \ell_2}}_{d \times d} \underbrace{K_{j,*}^\top}_{d \times 1}, \forall i \in [n], j \in [n].$$

We provide a claim and delay the proofs into Appendix (see Section C).

**Claim 4.6.** *Let $A^{\ell_1, \ell_2} \in \mathbb{R}^{n \times n}$ be defined as Definition 4.5. Then, we can show $A = \sum_{(\ell_1, \ell_2) \in S} A^{\ell_1, \ell_2}$.*

**Definition 4.7.** *Let $S$ be defined as in Definition 4.2. For each $(\ell_1, \ell_2) \in S$, we define matrix $C^{\ell_1, \ell_2} \in \mathbb{R}^{n \times n}$ as $C_{i,j}^{\ell_1, \ell_2} := (W_{i-j})_{\ell_1, \ell_2}$. This matrix is Toeplitz since $C_{i,j}^{\ell_1, \ell_2}$ depends only on $i - j$.*

We provide a claim and delay the proofs into Appendix (see Section C).

**Claim 4.8.** *Let $A^{\ell_1, \ell_2} \in \mathbb{R}^{n \times n}$ be defined as Definition 4.5. We can show*

$$A^{\ell_1, \ell_2} = \text{diag}(Q_{*, \ell_1}) C^{\ell_1, \ell_2} \text{diag}(K_{*, \ell_2}).$$

**Claim 4.9** (Running Time). *Let matrix $A^{\ell_1, \ell_2} \in \mathbb{R}^{n \times n}$ be defined as Definition 4.5. For any vector $x \in \mathbb{R}^n$, we can compute $A^{\ell_1, \ell_2} x$ in $O(n \log n)$ time using FFT.*

*Proof.* Using Claim 4.8, we can show that $A^{\ell_1, \ell_2}$ is a rescaled Toeplitz matrix. Thus, for any vector $v$, we can compute $A^{\ell_1, \ell_2} v$ in $O(n \log n)$ time. $\qquad\square$

## 5 CONCLUSION

In this work, we provide an almost linear time algorithm for RoPE attention. RoPE attention is used as a more expressive variant on attention in many applications, but the usual polynomial method approach inherently cannot work for calculating it quickly. We introduced a new way to combine the polynomial method with our "rescaled Toeplitz matrices" and the Fast Fourier transform in order to solve this problem more efficiently. As future work introduces more variants on attention, it will be exciting to explore whether these and other linear algebraic tools can still be used to perform fast computations.

ETHIC STATEMENT

This paper does not involve human subjects, personally identifiable data, or sensitive applications. We do not foresee direct ethical risks. We follow the ICLR Code of Ethics and affirm that all aspects of this research comply with the principles of fairness, transparency, and integrity.

REPRODUCIBILITY STATEMENT

We ensure reproducibility of our theoretical results by including all formal assumptions, definitions, and complete proofs in the appendix. The main text states each theorem clearly and refers to the detailed proofs. No external data or software is required.

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

# Appendix

**Roadmap.** In Section A, we introduce some theoretical foundations and other basic preliminaries. In Section B, we introduce some background and present our hardness result. In Section C, we provide missing proofs for linear case. In Section D, we explain how to handle the exp units and present the proofs of our main result. In Section E, we provide more related work. In Section F, we discuss the limitations of the paper. In Section G, we present the impact statement. In Section H, we discuss LLM usage.

## A    PRELIMINARIES

In this Section, we introduce concepts about nearly-linear time and almost-linear time.[2]

**Definition A.1.** *We say $O(n \operatorname{poly}(\log n))$ is nearly-linear time, and $O(n^{1+o(1)})$ is almost-linear time.*

Then we introduce the relationship between $O(n \log n)$ and $O(n^{1+o(1)})$.

**Fact A.2.** *We can show that $O(n \operatorname{poly}(\log n)) \leq O(n^{1+o(1)})$.*

*Proof.* First, observe that $\operatorname{poly}(\log n) = n^{O(\log(\log n))/\log n}$. Since $\log(\log n)/\log n \to 0$ as $n \to \infty$, we have that $\log(\log n)/\log n = o(1)$.

Therefore:

$$n \operatorname{poly}(\log n) = n \cdot n^{O(\log(\log n))/\log n}$$
$$= n^{1+O(\log(\log n))/\log n}$$
$$= n^{1+o(1)}$$

This directly shows that $O(n \operatorname{poly}(\log n)) \leq O(n^{1+o(1)})$. $\qquad\square$

## B    BACKGROUND ON HARDNESS AND COMPLEXITY

In Section B.1, we introduce some background and the low bound existence. In Section B.2, we present our hardness result.

### B.1    LOW BOUND EXISTENCE

In computational complexity theory, algorithmic hardness refers to the inherent difficulty of solving computational problems, measured by the resources (such as time and space) required for their resolution. As established in Garey and Johnson's foundational work "Computers and Intractability" (Garey & Johnson, 1990), understanding this hardness helps researchers and practitioners determine whether efficient solutions exist for given problems. Particularly significant in this context, lower bounds serve as a critical theoretical tool for establishing the minimum resources required to solve specific computational problems. This naturally leads us to examine how lower bounds play a fundamental role in computational complexity theory, establishing fundamental limits on the resources required to solve computational problems. As discussed in "Introduction to the Theory of Computation" (Sipser, 2006) (Chapter 9) and "Computational Complexity: A Modern Approach" (Arora & Barak, 2009) (Chapter 3), proving lower bounds helps us understand the inherent difficulty of problems and provides insights into computational hierarchies.

**Fact B.1** (Lower Bound Existence, (Sipser, 2006; Moore & Mertens, 2011)). *If the following holds:*

- *$\mathcal{A}$ is the set of all possible algorithms*

- *$\operatorname{Resources}(A)$ denotes the resource usage of algorithm $A$*

---

[2]We include this discussion into the paper due to the request from ICLR 2025 reviewer `https://openreview.net/forum?id=AozPzKE0oc`.

- *Succeeds$(A, P)$ indicates that algorithm $A$ correctly solves problem $P$*

- *$LB_{\mathcal{C}}$ represents the lower bound for class $\mathcal{C}$*

- *$f(n)$ is a function of the input size $n$*

*For proving computational complexity lower bounds, we can establish the following:*
*Let $\mathcal{C}$ be a class of computational problems. To prove that all algorithms solving problems in $\mathcal{C}$ require at least $f(n)$ resources (time or space), it is sufficient to demonstrate that there exists a single problem instance $P \in \mathcal{C}$ for which no algorithm using less than $f(n)$ resources can correctly solve $P$.*

*Formally:*

$$\forall \mathcal{C} \exists P \in \mathcal{C} : [\forall A \in \mathcal{A}, \text{Resources}(A) < f(n) \implies \neg\text{Succeeds}(A, P)] \implies \text{LB}_{\mathcal{C}} \geq f(n)$$

### B.2 Hardness

In this section, we show The Strong Exponential Time Hypothesis. Over 20 years ago, Impagliazzo and Paturi (Impagliazzo & Paturi, 2001) introduced The Strong Exponential Time Hypothesis (SETH). It is a stronger version of the P $\neq$ NP conjecture, which asserts that our current best SAT algorithms are roughly optimal:

**Hypothesis B.2** (Strong Exponential Time Hypothesis (SETH)). *For every $\epsilon > 0$ there is a positive integer $k \geq 3$ such that $k$-SAT on formulas with $n$ variables cannot be solved in $O(2^{(1-\epsilon)n})$ time, even by a randomized algorithm.*

SETH is a popular conjecture which has been used to prove fine-grained lower bounds for a wide variety algorithmic problems, as discussed in depth in the survey (Williams, 2018).

**Theorem B.3** (Restatement of Theorem 1.4). *Assuming* SETH*, for every $q > 0$, there are constants $C, C_a, C_b > 0$ such that: there is no $O(n^{2-q})$ time algorithm for the problem* ARAttC$(n, d = C \log n, B = C_b \sqrt{\log n}, \epsilon = n^{-C_a})$.

*Proof.* We will pick all of the $W_{-(n-1)}, \cdots, W_{(n-1)} \in \mathbb{R}^{d \times d}$ to be an identity $I_d$ matrix. Thus the RoPE attention becomes classical attention. Thus using (Alman & Song, 2023), our lower bound result follows. $\qquad\square$

## C    Missing Proofs for Linear Case

**Claim C.1** (Restatement of Claim 4.6). *Let $A^{\ell_1, \ell_2} \in \mathbb{R}^{n \times n}$ be defined as Definition 4.5. Then, we can show*

$$A = \sum_{(\ell_1, \ell_2) \in S} A^{\ell_1, \ell_2}.$$

*Proof.* For each $i \in [n], j \in [n]$, we compute each $(i, j)$-th entry of matrix $A \in \mathbb{R}^{n \times n}$ as

$$
\begin{aligned}
A_{i,j} &= Q_{i,*} W_{i-j} K_{j,*}^{\top} \\
&= Q_{i,*} \sum_{(\ell_1, \ell_2) \in S} W_{i-j}^{\ell_1, \ell_2} K_{j,*}^{\top} \\
&= \sum_{(\ell_1, \ell_2) \in S} Q_{i,*} W_{i-j}^{\ell_1, \ell_2} K_{j,*}^{\top} \\
&= \sum_{(\ell_1, \ell_2) \in S} A_{i,j}^{\ell_1, \ell_2}
\end{aligned}
$$

where the second step follows from Claim 4.4, the third step follows from rearranging the summation, and the last step follows from the definition of $A_{i,j}^{\ell_1, \ell_2}$.

Thus, we complete the proof. $\qquad\square$

**Claim C.2** (Restatement of Claim 4.8). *Let $A^{\ell_1,\ell_2} \in \mathbb{R}^{n \times n}$ be defined as Definition 4.5. We can show*

$$A^{\ell_1,\ell_2} = \mathrm{diag}(Q_{*,\ell_1})C^{\ell_1,\ell_2}\,\mathrm{diag}(K_{*,\ell_2}).$$

*Proof.* We can rewrite $A_{i,j}^{\ell_1,\ell_2}$ as follows

$$
\begin{aligned}
A_{i,j}^{\ell_1,\ell_2} &= Q_{i,*}W_{f(i-j)}^{\ell_1,\ell_2}K_{j,*}^\top \\
&= Q_{i,*}e_{\ell_1}(W_{f(i-j)})_{\ell_1,\ell_2}e_{\ell_2}^\top K_{j,*}^\top \\
&= Q_{i,\ell_1}(W_{f(i-j)})_{\ell_1,\ell_2}K_{j,\ell_2}
\end{aligned}
$$

We define $C_{i,j}^{\ell_1,\ell_2} = (W_{f(i-j)})_{\ell_1,\ell_2}$, then the above equation becomes

$$A_{i,j}^{\ell_1,\ell_2} = Q_{i,\ell_1}C_{i,j}^{\ell_1,\ell_2}K_{j,\ell_2}$$

Thus we can have

$$A^{\ell_1,\ell_2} = \mathrm{diag}(Q_{*,\ell_1})C^{\ell_1,\ell_2}\,\mathrm{diag}(K_{*,\ell_2})$$

Therefore, we complete the proof. $\square$

# D   HOW TO HANDLE THE EXP TERMS

We now give our full algorithm for general RoPE attention. In Section D.1, we study matrices which are the entry-wise products of a number of rescaled Toeplitz matrix, and how to use that decomposition to quickly multiply such matrices with a vector. In Section D.2, we show how to decompose the RoPE attention matrix into summation of a number of such structured matrices using the polynomial method. In Section D.3, we show how to put everything together to get our main result.

## D.1   THE RUNNING TIME OF HAMADARD PRODUCT OF RESCALED TOEPLITZ MATRIX MULTIPLYING A VECTOR

**Lemma D.1.** *Let $m : [d] \times [d] \to \mathbb{N}$ be any function[3]. Define the matrix $A^{(m)} \in \mathbb{R}^{n \times n}$ by*

$$A_{i,j}^{(m)} := \prod_{(\ell_1,\ell_2) \in S}(A_{i,j}^{\ell_1,\ell_2})^{m(\ell_1,\ell_2)}, \forall i \in [n], j \in [n].$$

*Then $A^{(m)}$ is also of the form of a rescaled Toeplitz matrix (see Definition 3.7). Furthermore, for any vector $v \in \mathbb{R}^n$, $A^{(m)}v$ can be computed in $O((\sum_{(\ell_1,\ell_2) \in S} m(\ell_1,\ell_2)) \cdot n \log n)$ time.[4]*

*Proof.* We define set $S$ to be

$$\{(\ell_{1,1},\ell_{2,1}),(\ell_{1,2},\ell_{2,2}),\cdots,(\ell_{1,|S|},\ell_{2,|S|})\} \subset [d] \times [d]$$

We define $t_i \in \mathbb{N}$ for each $i \in [|S|]$ as follows

$$t_i := m(\ell_{1,i},\ell_{2,i}).$$

From the definition of $A_{i,j}^{(m)} \in \mathbb{R}$, we know that $A^{(m)} \in \mathbb{R}^{n \times n}$ can be written as the entry-wise product of a collection of matrices (where each matrix is a rescaled Toeplitz matrix), i.e.,

$$A^{(m)} = (A^{\ell_{1,1},\ell_{2,1}})^{\circ t_1} \circ \cdots \circ (A^{\ell_{1,|S|},\ell_{2,|S|}})^{\circ t_{|S|}}$$

Using Lemma 3.9, we know the entry-wise product between any two rescaled Toeplitz matrix is still a rescaled Toeplitz matrix. Thus, applying Lemma 3.9 to the above equations for $\sum_{i=1}^{|S|} t_i$ times, we can show that $A^{(m)}$ is still a rescaled Toeplitz matrix.

Using Lemma 3.10, we know that for any vector $v$, $A^{(m)}v$ can be computed in $O((\sum_{i=1}^{|S|} t_i) \cdot n \log n)$ time. $\square$

---

[3]Here intuitively, $m$ represents the exponents of variables in a monomial of a polynomial.

[4]Later, we will show that $\sum_{(\ell_1,\ell_2) \in S} m(\ell_1,\ell_2) = n^{o(1)}$ for the function $m$ we used in this paper.

## D.2 EXPANDING POLYNOMIALS INTO SUMMATION OF SEVERAL RESCALED TOEPLITZ MATRICES

**Lemma D.2.** *Let $M^1, \ldots, M^k \in \mathbb{R}^{n \times n}$ be rescaled Toeplitz matrices. Let $p : \mathbb{R} \to \mathbb{R}$ be a polynomial of degree $\widetilde{d}$. Let $m \in \mathcal{M}$ be the set of functions $m : [k] \to \mathbb{N}$ such that $\sum_{\ell=1}^{k} m(\ell) \leq \widetilde{d}$. Consider the matrix $M \in \mathbb{R}^{n \times n}$ defined by $M_{i,j} := p(\sum_{\ell=1}^{k} M_{i,j}^{\ell})$. Then $M \in \mathbb{R}^{n \times n}$ can be written as the following sum of rescaled Toeplitz matrices:*

$$M = \sum_{m \in \mathcal{M}} \alpha_m \cdot N^{(m)}$$

*Here $N^{(m)} \in \mathbb{R}^{n \times n}$ is defined as $N_{i,j}^{(m)} = (M_{i,j}^{\ell})^{m(\ell)}$ for all $i \in [n], j \in [n]$ and $\alpha_m \in \mathbb{R}$ is coefficient. Furthermore, the number of rescaled Toeplitz matrices is $|\mathcal{M}| = O(\binom{\widetilde{d}+k}{k})$.*

*Proof.* Recall $\mathcal{M}$ is the set of functions $m : [k] \to \mathbb{N}$ such that $\sum_{\ell=1}^{k} m(\ell) \leq \widetilde{d}$. Then, for each $m \in \mathcal{M}$ there is a coefficient $\alpha_m \in \mathbb{R}$ such that we can rewrite polynomial $p$ as follows:

$$p(z_1 + \cdots + z_k) = \sum_{m \in \mathcal{M}} \alpha_m \cdot \prod_{\ell=1}^{k} z_\ell^{m(\ell)}. \tag{2}$$

Thus,

$$M_{i,j} = p(\sum_{\ell=1}^{k} M_{i,j}^{\ell})$$

$$= \sum_{m \in \mathcal{M}} \alpha_m \cdot \prod_{\ell=1}^{k} (M_{i,j}^{\ell})^{m(\ell)}$$

$$= \sum_{m \in \mathcal{M}} \alpha_m \cdot N^{(m)}$$

where the first step follows from definition of $M$, the second step follows from Eq. (2), and the last step follows from definition of $N^{(m)}$. Thus, we can see $M = \sum_{m \in \mathcal{M}} \alpha_m \cdot N^{(m)}$.

□

## D.3 MAIN RESULT

Finally, we are ready to put all our techniques together.

**Theorem D.3** (Restatement of Theorem 1.3)**.** *Suppose $d = O(\log n)$ and $B = o(\sqrt{\log n})$. There is an $n^{1+o(1)}$ time algorithm to approximate ARAttC up to $\epsilon = 1/\operatorname{poly}(n)$ additive error.*

*Proof.* We use the polynomial of Lemma 3.1 in Lemma D.2 with choice of $k = |S| = O(d) = O(\log n)$ and $\widetilde{d} = o(\log n)$ is the degree of the polynomial from Lemma 3.1 for error $1/\operatorname{poly}(n)$. We can thus upper bound

$$|\mathcal{M}| = O(\binom{k + \widetilde{d}}{\widetilde{d}}) = n^{o(1)}.$$

The total running time consists of three parts: first, approximating $A\mathbf{1}_n$ which gives an approximation to diagonal matrix $D$; second, approximating $Av$ for $d$ different columns vectors $v$, this will approximate $AV$; third, combining approximation of $D^{-1}$ with approximation of $AV$, to obtain an approximation of $D^{-1}AV$. Combining Lemma D.1 and D.2. The dominating running time for above three parts is

$$|\mathcal{M}| \cdot \sum_{(\ell_1, \ell_2) \in S} m(\ell_1, \ell_2) \cdot n \log n = O(n^{1+o(1)})$$

Due to the choice of $|\mathcal{M}| = n^{o(1)}, |S| = O(d), d = O(\log n)$.

The error analysis remains identical to prior attention algorithms using the polynomial method (Alman & Song, 2023), thus we omit the details here. □

## E MORE RELATED WORK

**Fast Fourier transform.** The Fast Fourier transform algorithm (Cooley & Tukey, 1965) can multiply the $n$ by $n$ Discrete Fourier transform matrix times an input vector in $O(n \log n)$ time. This algorithm is impactful in many areas, including image processing, audio processing, telecommunications, seismology, and polynomial multiplication. Due to its fundamental importance, a significant body of modern research has been dedicated to further accelerating the Fast Fourier transform. These efforts include decreasing the number of required arithmetic operations (Sergeev, 2017; Alman & Rao, 2023), reducing the sample complexity in the sparse setting (Candes & Tao, 2006; Rudelson & Vershynin, 2008; Blumensath & Davies, 2010; Needell & Vershynin, 2010; Bourgain, 2014; Haviv & Regev, 2017; Nakos et al., 2019), and improving the running time in the sparse setting (Gilbert et al., 2012; Hassanieh et al., 2012a;b; Indyk & Kapralov, 2014; Indyk et al., 2014; Price & Song, 2015; Moitra, 2015; Kapralov, 2016; 2017; Chen & Price, 2019b;a; Kapralov et al., 2019; Jin et al., 2023; Song et al., 2023a; Li et al., 2025b). Some recent studies (Yu et al., 2023; Ahmad et al., 2023; Gao et al., 2024b; Li et al., 2024b) have explored leveraging machine learning techniques to optimize FFT performance in practical scenarios. Other works have investigated hardware-specific optimizations to further enhance computational efficiency, particularly in large-scale applications.

## F LIMITATIONS

This work presents an almost linear-time algorithm for RoPE attention, supported by theoretical analysis. However, we do not include any empirical evaluations to validate the practical performance of the proposed method.

## G IMPACT STATEMENT

This work introduces the first almost linear-time algorithm for RoPE attention, providing a novel solution and new insights into the computational bottlenecks of RoPE-based attention mechanisms. It has the potential to accelerate future large language model (LLM) training and evaluation. As this is a purely theoretical contribution, we do not foresee any negative social impacts.

## H LLM USAGE DISCLOSURE

LLMs were used only to polish language, such as grammar and wording. These models did not contribute to idea creation or writing, and the authors take full responsibility for this paper's content.

