# OpenReview forum: "Fast RoPE Attention: Combining the Polynomial Method and Fast Fourier Transform"
_ICLR.cc/2026/Conference — Submitted to ICLR 2026_

### Official Review · Reviewer_3Gw1 · 2025-10-17

**Soundness:** 2
**Presentation:** 2
**Contribution:** 1
**Rating:** 2
**Confidence:** 4

**Summary:**

The paper presents a fast, almost linear-time algorithm for approximating RoPE-based attention. The main technique of the proposed method is a combination of the polynomial method and the Fast Fourier Transform (FFT) to handle the structure of RoPE. The paper also provides a computational hardness result.

**Strengths:**

RoPE is a cornerstone of many state-of-the-art LLMs (Llama, Claude, etc.), and developing faster algorithms for it is of significant practical interest.

**Weaknesses:**

1). The technical novelty is limited. The paper uses FFT to handle Toeplitz-like structures in positional encodings, which is also a known approach in existing models [1]. The primary contribution is the specific application of this technique to RoPE and combining it with the polynomial method which is yet another known method. While this is a valid contribution, the paper fails to articulate more generalizable algorithmic insight beyond this direct combination.

[1] Qin, Zhen, et al. "Toeplitz Neural Network for Sequence Modeling." The Eleventh International Conference on Learning Representations.

2). The assumptions in the theorems, $B=o(\sqrt{\log n})$ and $d= O(\log n)$, are not validated. In particular I doubt if $d= O(\log n)$ is true in practice. For current LLMs, the embedding size is very large. For example, the embedding size of DeepSeek-R1 is 7168, which is unlikely to be of $\log n$ scale.

3). Although the research problem is driven by practical applications, the paper is purely theoretical. Without experiments, it is impossible to assess:
  - The practical speed-up over standard, hardware-accelerated attention.
  - The actual trade-offs between speed and approximation error.
  - How the method compares to other approximate attention mechanisms in terms of quality and performance.
  - The overhead introduced by the FFT and polynomial coefficient computations.
As a result, the paper's claims of efficiency cannot be translated into practical innovations.

4).The related work section is weirdly long and diverges from the main topic. For example, it is not clear why accelerating diffusion models or GNNs is relevant to this work.

**Questions:**

The paper mentions that Claude uses RoPE. How do you know that?

---

> ### Author Response · Authors · 2025-11-17
> **Reply to Reviewer 3Gw1 - Part 1**
>
> Thank you for your thoughtful review. To address the weaknesses:
> * You’re right that both the FFT and polynomial method are very prominent algorithmic techniques in the literature, and have both been used for attention computation before. A main technical contribution of our paper is that we combine the two, which we believe has never been done before. We discuss in our technique overview (see especially lines 180-198 of our submission) why the polynomial method could not solve RoPE attention on its own, since the RoPE attention matrix provably does not have a low-rank approximation, whereas the polynomial method is a low-rank approximation technique. We needed to overcome this conceptual barrier and find a new way to use it in conjunction with the FFT. In particular, we believe that the fact that these two techniques are so commonly used, and yet had never been combined before our work, should be viewed as a _strength_ rather than as a weakness of our paper.
>
> For what it’s worth, the reference [1] you give does not seem to be using FFT in a way which is comparable with what we do in our work. We use FFT as part of our algorithm to approximate RoPE attention. They are not using FFT to compute or approximate any variant of attention. Rather, they give a different, purely position-driven token-mixing mechanism that uses FFT.
> * These regimes of B and d in terms of n are the most reasonable to focus on for two different reasons. First, these are the same regimes which are typically studied in the literature on the theory of attention. d = O(log n) is used in all the prior work on attention algorithms that we build on [1,2,3,4]. Moreover, this specific choice of d and B exactly matches the parameters of prior work on (non-RoPE) attention algorithms [1]. In general, when given specific numbers for d that are used (like 7168), it’s hard to tell whether or not it’s reasonable to model this as O(log n) since the O might be hiding any constant. However, the theory literature has consistently chosen d = O(log n) which most accurately captures the high-dimensionality of attention. Second, we prove that the parameters of our algorithmic result are tight, by giving an accompanying lower bound. If one is in a regime where B is too big and our algorithm does not apply, then our lower bound shows that no subquadratic algorithm is possible, using any conceivable method. In other words, the B we chose is optimal, and cannot possibly be improved.
> * The goal of our paper is to continue the recent line of work on determining theoretically whether fast algorithms are possible for attention. Establishing the theoretical limits of what is and is not possible for fast attention is critical to understanding how LLM systems can be improved, and directing future empirical work, and these works are typically published in ICLR and other top ML venues for this reason [1,2,3,4]. Similar to these prior works, we don’t include experiments here but expect that future empirical work will build on it, since establishing the theory is important for directing the practice. For example, here are a few prior empirical works that directly built on the previous theory results in this area: [5,6,7] In particular, since RoPE attention appears more complicated than standard attention, and prior techniques do not directly apply to it (see our technique overview on page 4), we believe it’s quite surprising that we were able to extend this line of work to RoPE.
> Regarding the overhead of the FFT: the FFT is perhaps one of the most practical and widely-applied algorithms, and has even been used before in empirical work on attention. For one recent example see [1].
>
> [1] Fast Attention Requires Bounded Entries. Josh Alman, Zhao Song. NeurIPS 2023.
>
> [2] Metric Transforms and Low Rank Representations of Kernels for Fast Attention. Timothy Chu, Josh Alman, Gary L. Miller, Shyam Narayanan, Mark Sellke, Zhao Song. NeurIPS 2024.
>
> [3] The Fine-Grained Complexity of Gradient Computation for Training Large Language Models. Josh Alman, Zhao Song. NeurIPS 2024.
>
> [4] How to Capture Higher-order Correlations? Generalizing Matrix Softmax Attention to Kronecker Computation. Josh Alman, Zhao Song. ICLR 2024.
>
> [5] H2O: Heavy-Hitter Oracle for Efficient Generative Inference of Large Language Models. Zhenyu Zhang, Ying Sheng, Tianyi Zhou, Tianlong Chen, Lianmin Zheng, Ruisi Cai, Zhao Song, Yuandong Tian, Christopher Ré, Clark W. Barrett, Zhangyang Wang, Beidi Chen. NeurIPS 2023.
>
> [6] HyperAttention: Long-context Attention in Near-Linear Time. Insu Han, Rajesh Jayaram, Amin Karbasi, Vahab Mirrokni, David Woodruff, Amir Zandieh. ICLR 2024.
>
> [7] The Hedgehog & the Porcupine: Expressive Linear Attentions with Softmax Mimicry. Michael Zhang, Kush Bhatia, Hermann Kumbong, Christopher Re. ICLR 2024.
>
> [8] SPECTRE: An FFT-Based Efficient Drop-In Replacement to Self-Attention for Long Contexts. Jacob Fein-Ashley, Neelesh Gupta, Rajgopal Kannan, Viktor Prasanna. arXiv 2502.18394

---

> ### Author Response · Authors · 2025-11-17
> **Replay to Reviewer 3Gw1 - Part 2**
>
> * Our goal of the related work section is to demonstrate that the topics we’re studying in this paper are broadly and popularly studied in the literature, both to emphasize the reach of our results, and to clarify that we are not the first to come up with some of these concepts. For example, for diffusion models, the point we are aiming to make is that (1) most diffusion models use transformer backbones, meaning our results apply to them, and (2) a line of prior work has studied theoretical guarantees on efficiently approximating diffusion models, which our work fits in to. We feel discussing these is important to explain the context of our results, but are happy to shorten these sections.
>
> Question:
> * Thank you for pointing that out; we will remove the reference to Claude using RoPE.

---

### Official Review · Reviewer_6Kug · 2025-10-30

**Soundness:** 3
**Presentation:** 3
**Contribution:** 3
**Rating:** 4
**Confidence:** 3

**Summary:**

The paper proposes the first almost linear-time algorithm for RoPE attention under bounded entries, matching known lower bounds. It introduces a generalized RoPE attention problem (ARAttC), proves an upper bound via a novel combo of the polynomial method with FFT on sums of rescaled Toeplitz matrices, and a SETH-based lower bound.

**Strengths:**

(1) The motivation of this paper is clear. The paper explains why classic polynomial-method low-rank arguments break under RoPE (Toeplitz-like structure rather than low rank) and why FFT is the right technique.

(2) The method incorporates the polynomial approximation and fast computation of FFTs.

**Weaknesses:**

(1) The theorems are asymptotic; it would help to expose the exact dependence on the polynomial degree and the number of rescaled-Toeplitz summands t after approximation. Here, n is the sequence length. In real applications, will it be approaching \infty? I think the real LLMs have a sliding window, and n is not very large, right?

(2) The paper does not conduct any experiments to show the improvement of the computation efficiency. I suggest including some experiments (even small synthetic ones) to compare the proposed method with Flashattention, etc.

**Questions:**

(1) Is the polynomial approximation to the exp function stable? Especially when we are using higher-order polynomials.

(2) How does the algorithm extend across multi-head attention and batching?

(3) How does the approach interact with causal masking and sliding-window attention often used with RoPE?

---

> ### Author Response · Authors · 2025-11-17
> **Reply to Reviewer 6Kug - Part 1**
>
> Thank you for your thoughtful review. To address the weaknesses:
> * Indeed our results are asymptotic (as are basically all results in the domain of theoretical computer science). For one thing, we emphasize that our lower bound result shows that our algorithm is tight: if one is in a parameter regime where our algorithm does not apply, then our lower bound shows that it is impossible to design a subquadratic algorithm for the problem, using any conceivable algorithmic technique. And there is no asymptotic gap between our upper bound (B = O(sqrt( log n))) and lower bound (B = omega(sqrt( log n))).
> You’re right that techniques like sliding window are used in practice to decrease n in attention, in part to address its quadratic scaling. However, in the setting of our algorithm, we are able to compute the entire RoPE attention in almost linear time, without needing something like sliding window! One could thus achieve the running time of sliding window while still incorporating the entire context into the RoPE attention, or one could combine with sliding window to even further reduce the running time (see the answer to your question below for a bit more detail).
> For what it’s worth, the exact polynomial degree and other dependencies you mention can be determined in a straightforward way from the proofs we give. Lemma 3.1 nails down the needed polynomial degree, even up to the leading constant and low-order terms. It is then simply input into a binomial coefficient in the proof of Theorem D.3. If one had an application in mind with exact choices of n, d, eps, etc, then one could calculate exactly the resulting inner dimension in the proof of Theorem D.3.
> * The goal of our paper is to continue the recent line of work on determining theoretically whether fast algorithms are possible for attention. Establishing the theoretical limits of what is and is not possible for fast attention is critical to understanding how LLM systems can be improved, and directing future empirical work, and these works are typically published in ICLR and other top ML venues for this reason [1,2,3,4]. Similar to these prior works, we don’t include experiments here but expect that future empirical work will build on it, since establishing the theory is important for directing the practice. For example, here are a few prior empirical works that directly built on the previous theory results in this area: [5,6,7] In particular, since RoPE attention appears more complicated than standard attention, and prior techniques do not directly apply to it (see our technique overview on page 4), we believe it’s quite surprising that we were able to extend this line of work to RoPE.
>
> [1] Fast Attention Requires Bounded Entries. Josh Alman, Zhao Song. NeurIPS 2023.
>
> [2] Metric Transforms and Low Rank Representations of Kernels for Fast Attention. Timothy Chu, Josh Alman, Gary L. Miller, Shyam Narayanan, Mark Sellke, Zhao Song. NeurIPS 2024.
>
> [3] The Fine-Grained Complexity of Gradient Computation for Training Large Language Models. Josh Alman, Zhao Song. NeurIPS 2024.
>
> [4] How to Capture Higher-order Correlations? Generalizing Matrix Softmax Attention to Kronecker Computation. Josh Alman, Zhao Song. ICLR 2024.
>
> [5] H2O: Heavy-Hitter Oracle for Efficient Generative Inference of Large Language Models. Zhenyu Zhang, Ying Sheng, Tianyi Zhou, Tianlong Chen, Lianmin Zheng, Ruisi Cai, Zhao Song, Yuandong Tian, Christopher Ré, Clark W. Barrett, Zhangyang Wang, Beidi Chen. NeurIPS 2023.
>
> [6] HyperAttention: Long-context Attention in Near-Linear Time. Insu Han, Rajesh Jayaram, Amin Karbasi, Vahab Mirrokni, David Woodruff, Amir Zandieh. ICLR 2024.
>
> [7] The Hedgehog & the Porcupine: Expressive Linear Attentions with Softmax Mimicry. Michael Zhang, Kush Bhatia, Hermann Kumbong, Christopher Re. ICLR 2024.

---

> ### Author Response · Authors · 2025-11-17
> **Reply to Reviewer 6Kug - Part 2**
>
> Questions:
> * Yes, the cited result for our Lemma 3.1 shows that the polynomial has small coefficients which won’t substantially impact the numerical instability. We’re not sure exactly what you mean by “higher-order polynomial”, but the polynomial we use has very small (n^{o(1)}) degree, and it actually comes from plugging in an inner product into a single-variable polynomial for the exponential, so that few alternations or cancellations need to happen.
> * With multiple heads/layers/batching/etc, the algorithm can be applied separately to each head and combined together. The matrix-vector multiplications of different heads can be done in parallel, and the incurred error is so small (1/poly(n)) that they do not accumulate too much over an entire network.
> * There are many standard ways in which algorithms for (usual) attention can be used for sliding window/causal masking/etc (such as partitioning the parts of the attention matrix that are in the masking into block matrices, and using the algorithm on each block), and these approaches can be used here in the RoPE setting as well. That said, perhaps an exciting consequence of our work is that it runs in almost linear time without needing something like sliding window; our algorithm runs in the same amount of time and considers all pairwise interactions.

---

### Official Review · Reviewer_cCXb · 2025-10-30

**Soundness:** 3
**Presentation:** 2
**Contribution:** 3
**Rating:** 6
**Confidence:** 2

**Summary:**

This paper addresses a polynomial methods for efficient computation of attention, with **Rotary Position Embeddings** (RoPE).
While prior works achieved near-linear algorithms for standard attention(under bounded entry regimes), these methods do not extend to the increasingly popular RoPE variant.
The authors develop a new algorithm that achieves near-linear time for RoPE attention computation, combining the polynomial method (for low-rank approximations) with the Fast Fourier Transform
using rescaled Toeplitz formulation.
 They prove that the nearly linear regime's theoretical thresholds extend to RoPE and provide the first such provable algorithms for this popular class of attention mechanisms.

**Strengths:**

Clear Motivation & Relevance: The paper highlights the importance of efficient attention mechanisms, especially as RoPE becomes standard in large LLMs (Llama, Claude, Gemini, Apple, etc.).

Originality: The combination of the polynomial method with FFT for rescaled Toeplitz matrices is novel, and the authors identify why previous techniques fail in the RoPE case.

Theoretical Rigor: Strong upper and lower bounds are established. The authors are meticulous in showing tightness of their results, connecting them with SETH and prior complexity theory for attention.

Exposition: Section structure is logical and easy to follow. Notation, background, and step-wise algorithmic development are mostly clear.
The proofs and more technical details are referenced for reproducibility.

**Weaknesses:**

Clarity: Some sections (esp. regarding structured matrix manipulations) assume a degree of background with FFT applications and polynomial approximations in algorithms.
 Additional diagrams or simplified intuition would make the work more accessible to a wider ML/AI audience.

Related Work Scope: The related work is comprehensive regarding theoretical literature, but more discussion about current practical/engineering solutions for fast attention
(e.g., FlashAttention variants, hardware-accelerated solutions) might provide context for potential synergy or limitations.
Also, it is not clear if the RoPE issue is also present in Linear attention methods, where the attention is represented by matrix multiplication of (Mercer's) kernel functions.

Generality: The main result holds under bounded norm assumptions and for certain embedding sizes relative to sequence length (O(log n)). While justified, practical consequences
 and possible relaxations/tighter practical bounds could be discussed more. Can you point in which cases this bound breaks? is it model/ data dependent?

Experimental Results Are Missing: Up to page 9 (end of main content), the paper is entirely theoretical.
 There is no validation of the algorithm's practical performance on real-world attention problems or large models (e.g., LLMs using RoPE in practice).
 While the theory is strong, empirical evidence for efficiency, accuracy, and scalability trade-offs is necessary for broader impact.

**Questions:**

$Q_1$. Precision/Accuracy Robustness: How does the practical choice of polynomial approximation error $\epsilon$ and norm bound $B$ affect downstream accuracy and speed?
 Are there scenarios (e.g., quantized or noisy activations) where theoretical advantages may not materialize?

$Q_2$. Scalability: while the paper highlights regime restrictions (O(logn)), is it possible to extend your approach to non-logarithmic scenarios in practice? or will a break in theoretical guarantees?

$Q_3$. Hardware implications: Given the rising importance of custom hw accelerators, is your method amenable to efficient hardware implementation/composability with existing frameworks?

$Q_4$. Empirical Validation: Do you plan to provide experiments on LLM inference/training with RoPE using your algorithm?
 How does the runtime and accuracy compare to current best practical methods (e.g., FlashAttention) on models like Llama-2/3?

$Q_5$. Linear attention extention: A vast class of fast attention methods are linear attention. Does the RoPE issue present also in this class of attentions? Does your method applicable to this case (e.g., Nystrom, Performer, etc.)?

$Q_6$. Low-dregree polynomial approximation: In lines 181-190 the paragraph explains that exp function can be approximated by a polynomial function.
 However, softmax defintion is diffferent and will have a different approximation error when using low degree polynomial to apprixmate.
 You should clarify the consequences of low-degree approximation of softmax rather than exp.

$Q_7$. Minor comments :
 - definition in line 63, is repeating again in line 68.
 - in line 60 "this lower bound" refers to ?
 - equation (1) is per head (should be clarified)
 - line 134, "changing the many parameters" - > "changing many parameters"
 - line 138, norm |S| should be defined (or referenced).
 - line 151, typo ==> 1/sqrt(d) (this sqrt{d} term should appear also in eq (1))

---

> ### Author Response · Authors · 2025-11-17
> **Reply to Reviewer cCXb - Part 1**
>
> Thank you for your thoughtful review. To address the weaknesses:
> * Clarity: Thank you for the feedback. We tried to cite the relevant background in the preliminaries section, for instance Fact 3.3 explains how to use the FFT for circulant matrices and cites the detailed review by (Gray et al) for the unfamiliar reader. We aimed to give a high-level overview of the whole approach in the technique overview (on lines 200-222). Are there any specific proofs where it would help for us to expand more?
> * Related Work: We aimed to focus our related work section on the background most relevant to what we’re doing here because of space constraints, but you’re absolutely right that there’s a potential for synergy with other approaches to fast attention which would be exciting to explore.  RoPE actually works poorly in conjunction with linear attention for the same reason we outline in lines 180-199: the RoPE part of the attention matrix is not low-rank, and really defined an n by n full rank matrix.
> * Generality: We aimed to focus on the parameter regimes (like d = O(log n)) which are most often studied in the literature on the theory of attention algorithms. But we particularly want to emphasize that our lower bound result shows that our algorithm is tight: if one is in a parameter regime where our algorithm does not apply, then our lower bound shows that it is impossible to design a subquadratic algorithm for the problem, using any conceivable algorithmic technique. We describe in lines 75-83 many examples where bounded entries arise in practice, although there are of course many examples of very large entries being used where our algorithm would not apply.
> * Experimental Results: The goal of our paper is to continue the recent line of work on determining theoretically whether fast algorithms are possible for attention. Establishing the theoretical limits of what is and is not possible for fast attention is critical to understanding how LLM systems can be improved, and directing future empirical work, and these works are typically published in ICLR and other top ML venues for this reason [1,2,3,4]. Similar to these prior works, we don’t include experiments here but expect that future empirical work will build on it, since establishing the theory is important for directing the practice. For example, here are a few prior empirical works that directly built on the previous theory results in this area: [5,6,7] In particular, since RoPE attention appears more complicated than standard attention, and prior techniques do not directly apply to it (see our technique overview on page 4), we believe it’s quite surprising that we were able to extend this line of work to RoPE.
>
> [1] Fast Attention Requires Bounded Entries. Josh Alman, Zhao Song. NeurIPS 2023.
>
> [2] Metric Transforms and Low Rank Representations of Kernels for Fast Attention. Timothy Chu, Josh Alman, Gary L. Miller, Shyam Narayanan, Mark Sellke, Zhao Song. NeurIPS 2024.
>
> [3] The Fine-Grained Complexity of Gradient Computation for Training Large Language Models. Josh Alman, Zhao Song. NeurIPS 2024.
>
> [4] How to Capture Higher-order Correlations? Generalizing Matrix Softmax Attention to Kronecker Computation. Josh Alman, Zhao Song. ICLR 2024.
>
> [5] H2O: Heavy-Hitter Oracle for Efficient Generative Inference of Large Language Models. Zhenyu Zhang, Ying Sheng, Tianyi Zhou, Tianlong Chen, Lianmin Zheng, Ruisi Cai, Zhao Song, Yuandong Tian, Christopher Ré, Clark W. Barrett, Zhangyang Wang, Beidi Chen. NeurIPS 2023.
>
> [6] HyperAttention: Long-context Attention in Near-Linear Time. Insu Han, Rajesh Jayaram, Amin Karbasi, Vahab Mirrokni, David Woodruff, Amir Zandieh. ICLR 2024.
>
> [7] The Hedgehog & the Porcupine: Expressive Linear Attentions with Softmax Mimicry. Michael Zhang, Kush Bhatia, Hermann Kumbong, Christopher Re. ICLR 2024.

---

> ### Author Response · Authors · 2025-11-17
> **Reply to Reviewer cCXb - Part 2**
>
> Thank you for the great questions. Many of these would be exciting directions for future work, although we attempt to give answers here:
> * Q1, in short, the error of the polynomial you pick determines the error at each attention head, and then these errors will accumulate in a straightforward way over an entire network. One could always pick a polynomial of high enough degree to make the error small; we focus on 1/poly(n) error in the paper since this is small enough to dwarf all the error accumulation in a network.
> * Q2, yes one could extend to other parameter regimes, although d = O(log n) is the regime most often considered in practice to more faithfully capture the parameter dynamics in the problem. For example, for smaller d = o(log n), there are technically algorithms for many of these problems running in time n*2^d = n^{1+o(1)}, but these are wildly impractical when d is a very large constant, so theorists usually do not consider this regime.
> * Q3 That’s a great question for follow-up work. Given that both polynomial/kernel methods and the FFT have been explored in work using hardware acceleration, we are optimistic that this could be beneficial here as well.
> * Q4, This is an exciting future direction, which we hope will be the subject of many follow-up works. For instance, above we cited multiple empirical follow-ups [5,6,7] to the polynomial method algorithm for (non-RoPE) attention [1].
> * Q5, It is not straightforward to use RoPE in conjunction with linear attention, since as discussed above, RoPE results in the attention matrix having full rank (n). However, our technique here could be used in conjunction with linear attention: one could use FFT for the RoPE part and linear attention techniques for the rest. This would correspond to our algorithm where we pick the polynomial p(x) = x to be the degree-1 identity function.
> * Q6 Using the polynomial method, approximating the softmax function and exp function is nearly identical. We ultimately use our approximation of exp to get a small error for both the numerator and denominator of softmax, and thus a good approximation of softmax. (This is why our algorithm works in two phases, where the first phase approximates D, the matrix of softmax denominators.)
> * Q7 Thank you for pointing these out; we will address them in the final version.

---

### Meta-Review · Area_Chair_Cc5x · 2026-01-08

**Summary:**

The paper examines if even in theory are there efficient algorithms for attention computation which can work with ROPE attention. They derive an almost linear time algorithm by combing previously used polynomial method and FFT. There were many concerns, majorly the following 1. The novelty of the techniques can be considered modest. 2. Lack of empirical validation.
As seen in the reviews, there is no strong support for the paper, and hence at this point it would be difficult to make a case for acceptance.

**Reviewer Concerns:**

Both concerns mentioned earlier are not addressed.

**Reviewer Scores:**

No one possibly.

---

### Decision · Program_Chairs · 2026-01-26

Reject